# Exposure to Amosite-Containing Ceiling Boards in a Public School in Switzerland: A Case Study

**DOI:** 10.3390/ijerph16245069

**Published:** 2019-12-12

**Authors:** David Vernez, Olivier Duperrex, Horacio Herrera, Vincent Perret, Isabelle Rossi, Frederic Regamey, Michel Guillemin

**Affiliations:** 1Center for Primary Care and Public Health (Unisanté), University of Lausanne, 1066 Epalinges, Switzerland; hh@hthh.ch (H.H.); Frederic.Regamey@unisante.ch (F.R.); 2Head of school health services, Unité PSPS, Canton of Vaud, 1014 Lausanne, Switzerland; Olivier.Duperrex@avasad.ch; 3Institute of Global Health, University of Geneva, 1202 Geneva, Switzerland; 4TOXpro SA, 1227 Carouge, Switzerland; vincent.perret@toxpro.ch; 5Service of Public Health, Department of Health and Social Action, Canton of Vaud, 1014 Lausanne, Switzerland; isabelle.rossi@vd.ch; 6Professor Emeritus, University of Lausanne, 1014 Lausanne, Switzerland; michel.guillemin@gmail.com

**Keywords:** asbestos, amosite, ceiling boards, school, health risk assessment

## Abstract

The measurement of an airborne concentration in Amosite fibers above 5035 F/m^3^ in a school prompted a retrospective quantitative health risk assessment. Dose estimates were built using air measurements, laboratory experiments, previous exposure data, and interviews. A dose response model was adapted for amosite-only exposure and adjusted for the life expectancy and lung cancer incidence in the Swiss population. The average yearly concentrations found were 52–320 F/m^3^. The high concentration previously observed was not representative of the average exposure in the building. Overall, the risk estimates for the different populations of the school were low and in the range of 2 × 10^−6^ to 3 × 10^−5^ for mesothelioma and 4 × 10^−7^ to 8 × 10^−6^ for lung cancer. The results evidenced however that children have to be considered at higher risk when exposed to asbestos, and that the current reference method and target values are of limited use for amphibole-only exposures. This study confirmed that quantitative health risk assessments and participatory approaches are powerful tools to support public decisions and build constructive communication between exposed people, experts, and policy-makers.

## 1. Introduction

### 1.1. General

The thermal resistance and flame retardant properties of asbestos, a naturally occurring mineral silicate fibre, made it an ideal insulation material. Its use spread from the early twentieth century, initially in high temperature processes and later in the building sector. Global asbestos production reached its peak in the early 1980s (4.7 million metric tonnes) [1]. The controversy surrounding the toxicity of the asbestos fibers eventually put an end to their usage in most industrialized countries, where production dropped in the 1980s and disappeared in the early 1990s.

There are two main mineralogical families of asbestos: amphiboles and serpentines. Amosite is a form of asbestos in the amphiboles family, characterized by straight fibrils and a diameter 3–10 times larger than chrysotile. All forms of asbestos are considered as carcinogenic and have been classified as such by the International Agency for Research on Cancer (IARC) since 1973 [2]—their potency differs and it is generally accepted that Amphiboles fibres exhibits a higher toxicity than serpentine fibres. Exposure to asbestos is the cause of several specific lung pathologies such as mesothelioma, asbestosis, or pleural plaques. It also contributes to the occurrence of non-specific pathologies, such as lung cancer. In Switzerland, the current annual incidence of mesothelioma in the Swiss population is of 0.6 cases/100,000 people in women and 4.1 cases/100,000 people in men [3].

Massive asbestos exposures were found historically in the mining and manufacturing sectors (e.g., asbestos textile industry). The available epidemiological evidence and dose-response relationships are based on cohorts among these highly exposed workers [4]. While these massive exposures disappeared in most industrialized countries, asbestos remains a major public and occupational health issue nowadays. Because of the ubiquitous nature of asbestos-containing materials, occupational exposure still occurs in sectors such as the construction trades (transformation, renovation or demolition) or in the waste sector. Despite the widespread ban on asbestos, the number of workers still exposed in European countries in 1990–1993 was estimated at 1,200,000 [5]. Moreover, workers in non-exposed trades or individuals from the general population could be exposed passively when occupying buildings containing asbestos materials.

### 1.2. Context of the Study

A situation of passive exposure to asbestos in a Swiss School is investigated in this study. The school complex is constituted of a main building, built in two stages (one in 1972 and the 1976), and a gymnasium, built in 1972. The presence of asbestos material in the building was known since 2005. Amosite fibers (about 1% in mass) were found in four out of 14 samples of the insulation boards lining the ceiling in the classrooms and corridors of the main building (the part built in 1972). No asbestos fibres other than amosite were found in the ceiling board. Because of the different construction periods and progressive replacement of the damaged parts, the distribution of the ceiling boards was inhomogeneous. Further investigations showed that the asbestos containing boards were located in the oldest part of the main building, which included about half of the 29 classrooms and covered slightly less than 50% of the total ceiling surface. Ceiling boards are considered as friable materials and are susceptible to release asbestos fibers when disturbed. Previous cases of asbestos-related diseases in relation with passive exposure to asbestos insulation material in building, including deteriorated acoustic insulation boards, have been reported [6]. For reasons unknown to the authors, the ceiling plates remained in place until recently.

In April 2015, following a detailed diagnosis of the building in the perspective of a forthcoming renovation, air measurements were carried out in several classes of the main building. Concentrations of 5035 F/m^3^ (fibers/m^3^) of Amosite asbestos fibers were observed in one classroom. Concentrations of 95–731 F/m^3^ were also observed in neighbouring rooms (*n* = 4) at the same period. In Switzerland, the 8-h OEL (occupational exposure limit) for asbestos is 0.01 F/mL (10,000 F/m^3^). A minimization principle is recommended for workers exposed passively, such as individuals working in asbestos-containing buildings material. The target value for passive exposure is of 0.001 F/mL (1000 F/m^3^). The same value is used for the general population [7]. The measurements of airborne asbestos concentrations five times above the thresholds tolerated in the general population and the closure of the classroom raised serious concerns among the building occupants and the pupil’s parents. These concerns were exacerbated by a seven-month delay between the latest air measurements and the closure of the classroom.

### 1.3. Aims

A quantitative health risk assessment was conducted, retrospectively, in order to assess the risk induced by the amosite-containing ceiling boards for the regular occupants of the building. Quantitative health risk assessments are increasingly used in the field of public health (e.g., air pollution) and allow policymakers to devise and implement more effective policies at local, national and global level. This approach was intended to bring impartial information to the teachers and the pupils’ parents in a sensitive communication context. More broadly, this study also presented the opportunity to address the relevance of the current standards and practices to assess similar indoor exposure situations.

## 2. Materials and Methods

### 2.1. Framework

The retrospective health risk assessment of the exposure to asbestos in the school was conducted by a joint working group, including external experts (hygienist, toxicologist, and occupational physician), internal experts from local authorities (public health, school health), and representatives of the exposed population (pupils’ parents and teachers). In this participative model, the pupils and teachers’ representatives had access to the same documentation than the external expert and were actively involved in the risk assessment process (e.g., data collection and analysis). Because of the specific context, involving schoolchildren and potentially elevated exposures to Amosite asbestos, a strong emphasis was given on the transparency of the expertise process. Additionally to their role in the working group, these representatives were also tasked to communicate the expertise progress to their peers. After each session of the working group, a summary of the progress of the work was publicly posted within the school for teachers, parents and students.

The risk assessment was focused on lung cancer and mesothelioma related to exposure to amosite. While exposure to asbestos is associated with various diseases, including also asbestosis, laryngeal cancer or ovarian cancer, it is generally considered that lung cancer and mesothelioma are the most sensitive effects. Exposure levels leading to acceptable risks for these two cancers were therefore considered as sufficiently protective for other forms of disease.

### 2.2. Study Population

Three populations, routinely frequenting the asbestos containing part of the building, were considered: the pupils, the teachers and the janitor of the school. The exposure of the janitor was considered as representative of the worst-case situation among the technical personnel of the school (e.g., cleaning workers) because of his daily presence in the building and proximity to the asbestos sources. Since 1972, the school welcomed pupils between six and 15 years old (primary school). It is estimated that over 3000 female and male pupils passed through the school between 1972 and 2015, although their exact number is not known. The evolution of school curricula in terms of weekly hours and school periods has been precisely documented since the 1960s (statistics from the department of education). It was therefore possible to determine the number of hours of school attendance over the while period. The cumulative duration spent in the school varies between 8000 and 11,000 h, depending on the student’s year of birth. An internal survey among the teachers (*n* = 47/77, rate of response 61%) indicated an average seniority in the school of nine years. The maximum seniority in the establishment was 13 years. With regard to the number of periods worked, the annual attendance corresponds to 582 h (average number of weekly hours, 38 weeks/year) and 1482 h (max. number of weekly hours, 38 weeks/year) for the medium and pessimistic scenario respectively.

### 2.3. Exposure Scenarios

Two sources of exposures were considered in this study: ambient exposure, which depends on the time spent into the building and the background indoor concentration, and event-related exposures, which are associated with specific events involving the asbestos-containing boards (e.g., the fall of a board into a classroom), the latter being dependent on the event frequency, duration, and exposure intensity.

Possible events were first identified by the working group through brainstorming and substantiated with interviews with the janitor, the technical service of the municipality and feedback from the teachers and pupils’ parents. In a second step, the frequency and duration of each event were estimated using various data sources and/or expert judgment. Average exposure scenarios and pessimistic scenarios were built for each population. Conservative mean values were used for the average scenario, while upper values were used in the pessimistic approach.

### 2.4. Exposure Concentrations

Ambient exposure concentrations and event-related exposure concentrations were estimated using air measurements whenever possible. Alternative methods were also considered when measurement was not possible or unlikely to produce consistent results with a reasonable number of samples. Detailed information about the exposure assessment methods used is available in Appendix A, in short the following methods were used:In situ measurements, according to the German norm VDI 3492:2013, were performed to assess ambient concentrations, and events-related concentrations associated with regular maintenance activities (e.g., changing a neon lamp in the ceiling board).Laboratory measurements were performed in an 8 m^3^ enclosed booth to assess events-related concentrations of potentially high emission (e.g., the fall or breakdown of a ceiling board). Concentrations in the classrooms were estimated, using IH Mod tool version 0.212 [8], and assuming a one-box dilution model.Previous technical data, issued from the Evalutil database [9] or reports issued from institutional bodies, was used when the two first methods were inappropriate.Monte-Carlo simulations have been conducted, using Stata IC 14.2 (StataCorp LP, 4905 Lakeway Drive College Station, TX, USA) in order to assess the maximal background concentration.

### 2.5. Dose-Response Model

The most common dose response model is derived from the Environmental Protection Agency model and Hodgson’s calculations on low dose exposures to chrysotile asbestos [10,11]. In this model, the concentration of exposure to fibers *f* (fibers/mL) is exponentially related to the incidence of mortality due to mesothelioma *I_M_* (rate/year). It takes into account the exposure duration *d* (years), the time after the first exposure *t* (years) and includes a multiplicative factor *K_M_*, determined by the carcinogenic potential of the pollutant.

(1)$$I_{M}=0,\;\mathrm{for}\;t<10$$

(2)$$I_{M}=K_{M}\cdot f\cdot {\left(t-10\right)}^{3},\;\mathrm{for}\;10+d>t>10,$$

(3)$$I_{M}=K_{M}\cdot f\cdot \left[{\left(t-10\right)}^{3}-{\left(t-10-d\right)}^{3}\right],\;\mathrm{for}\;t>10+d,$$

Considering the limits of the initial Environmental Protection Agency (EPA) model [12], the dose–response model used in this study for mesothelioma is the EPA model modified by the Dutch Expert Committee on Occupational Safety (DECOS) [12], taking into account the data adjustment proposed by Berman [4]. Two further adjustments were made to fit with the context of the study:The use of a specific *K_M_* value for Amosite rather than a unique value for amphiboles. A *K_M_* of 3.9 × 10^−8^ (95% CI: 2.6 × 10^−8^, 5.7 × 10^−8^), derived from the cohort of Seidman [13] and further adjusted by Berman [4], was considered.The risk calculation has been adjusted taking into account the life expectancy in the Swiss population [3]. As the incidence of I_M_ tends to increase with age and the pathology often manifests itself late, the life expectancy of the population concerned plays an important role in calculating the whole-life risk.

The dose–response model considered for lung cancers is based on the study of Stayner [14], who observed that the incidence of lung cancer *I_L_* is directly proportional to the exposure concentration f [fibers/mL], the duration of the exposure [years], and a multiplicative factor *K_L,_* reflecting the carcinogenic potential of the pollutant.

(4)$$I_{L}=K_{L}\cdot f\cdot d,$$

A *K_L_* of 1 × 10^−2^, based on available data at the time, was set by the EPA [10] and the World Health Organization [15]. A recent re-analysis of the data conducted by the DECOS, led to a *K_L_* 1.64 × 10^−2^ (95% CI: 0.34, 2.95). The *K_L_* value proposed by the DECOS was used in this study. The risk calculation has been further adjusted to take into account the life expectancy and the incidence of lung cancer in the Swiss population [3]. The code of the DECOS model, kindly provided by its authors, was adapted to the context of this study using R (ver. 3.2.4, The R Foundation for Statistical Computing, Vienna, Austria), and the packages data. table (ver. 1.9.6, The R Foundation for Statistical Computing, Vienna, Austria) and ggplot2 (ver. 2.1.0, The R Foundation for Statistical Computing, Vienna, Austria).

## 3. Results

### 3.1. Exposure Scenarios

Exposure scenarios for the three populations are presented in Table 1. Average and pessimistic estimates of the exposure duration for pupils and teachers in the building are of 9–11 years and 6–29 years, respectively. The school welcomes children during primary and secondary years. Each child residing in the neighbourhood is therefore expected to spend nine years in the building. Only one full-time janitor is working in the building. The average janitor’s exposure duration of 15 years is therefore de facto the true value. A full work-life exposure of 40 years was considered in the pessimistic scenario to reflect exposures of the previous janitors.

### 3.2. Exposures Concentrations

The frequency, duration and concentrations corresponding to each exposure situation considered as relevant by the working group are summarized in Table 2. Two distinct types of exposure sources were considered: (1) background noise, which comes from regular building use (jolts related to slamming doors, building vibrations, blinds use, etc.). This comes from very frequent and expected events and is therefore considered as a constant emission. (2) “Event-related” emissions, that result from a direct action on the boards, either expected or accidental (e.g., hitting a board). These event are less frequent and can cause occasional emission peaks. The total exposure is the sum of these components. In the average scenario, the exposure frequency is adjusted to take into account the fact that 50% of the ceiling boards contained asbestos. In the pessimistic scenario, it was assumed that each event involved asbestos-containing boards.

### 3.3. Background Exposure

Sixteen air measurements were performed (according to VDI 3492:2013) during the regular use of the building and in simulated use (with slamming of doors, closure of blinds, etc.). All 16 measurements were negative, suggesting a background concentration significantly lower than the Limit of Detection (LOD), of 95 and 190 F/m^3^, corresponding to an 8 h and 4 h sampling period respectively. As concentrations below 10^2^ F/m^3^ in Amosite fibres could be relevant for mesothelioma [12], an estimate of the maximal average background concentration was necessary. Monte Carlo simulation was used to predict a reasonable upper value for the geometric mean (GM) of the concentration distribution. It was found that, for a GM of 75 F/m^3^, the likelihood to get a negative result over the 16 samples taken was of 95% (details available in Appendix A). Interestingly, this concentration is comparable to levels found in schools in the UK equipped with asbestos-containing ceiling boards [16].

### 3.4. Events-Based Exposures

The ceiling boards are accessible to the occupants of the building. Occasional hits (e.g., throwing of objects, fixing a decoration), sufficient to induce small movements of the boards and potentially release of fibers, are expected. The maximal average concentration observed in situ when hitting repeatedly a board with a bracket (4 h, *n* = 2) was of 991 F/m^3^. A concentration of 1000 F/m^3^ and a frequency of one event per week and per classroom was considered.

The ceiling boards are also occasionally replaced when worn, broken or damaged by water. The replacement of a board could lead to a direct exposure of the janitor of the building as well as a passive exposure of the pupils and the teachers that will occupy the classroom during the next period. An exposure concentration of 50,000 F/m^3^ was estimated for the janitor. The passive exposure of the children during the 4 h following the board replacement was estimated to 400 F/m^3^. About five boards are replaced per year in the building.

The regular maintenance of the building includes the replacement of the neon lights on the ceiling. As the lights are in contact with the ceiling boards, this operation could lead to the release of fibers either due to the unintentional displacement of a board or to the release of the dust accumulated over the light casing. An exposure concentration of 20,000 F/m^3^, was estimated for the janitor. An exposure concentration of 200 F/m^3^, obtained through in situ measurements (maximal average value of 196 F/m^3^, 4 h, *n* = 4) was considered for the passive exposure in the classroom. A total of 75 lamp replacements, of about 10 min each, took place in the building every year.

Laboratory measurements suggested that the fall of a board (with or without board breaking) could lead to an exposure concentration of about 10,500 F/m^3^ in the classroom (maximal average value, *n* = 2, 1 h). The breaking of a board (without falling) leads to an exposure concentration in the classroom of 5500 F/m^3^.

In 2008, one of the classrooms was damaged by a fire. The remediation work probably led to a passive exposure of the building occupants during the following days. In the absence of adequate metrological data, a concentration of 10,000 F/m^3^ in the classroom (and 1000 F/m^3^ in the neighboring part of the building) during the remediation work was considered by the working group. This concentration was similar to the one observed during the fall and breaking of a board. This was judged sufficiently conservative, taking into account the relative small amount of asbestos-containing material and the fact that the wetting of the boards during the firefighting was lessening their emission potential.

### 3.5. Risk Assessment

Exposure concentration (Table 2) associated with asbestos-related events were combined to compute average yearly exposures for the different populations of the school. Along with the age of onset of exposure and the exposure duration (Table 1), this information was used to estimate yearly incidence rate of mortality for lung cancer I_L_ and mesothelioma I_M_, according to the dose model depicted in Section 2.4 (Equations (1)–(4)). Incidence rates are used to compute a number of yearly cases, considering the size of the remaining population at a given year (according to the life expectancy curve of the reference population). Whole-life risk is computed through the accumulation of expected yearly cases

The results, shown in Table 3 and Figure 1, are expressed in excess risk (ER) whole-life. The ER is the difference between the risk of disease in exposed vs. non-exposed subjects. It expresses in a population exposed to asbestos the risk of dying of mesothelioma or lung cancer throughout their lives. A risk of 1 × 10^−6^, means for example that exposure leads to one additional case of deaths in a population of 1,000,000 people exposed.

Using the adjusted dose-response model, it is possible to calculate the equivalent concentration, corresponding to a target excess risk and a given exposure scenario. The equivalent concentration, expressed in F/m^3^, is the average yearly exposure concentration that would lead to a specific ER level in a given exposure scenario (in terms of exposure duration). Equivalent concentrations were computed for some typical excess risks values and different types of asbestos fibers, using the Swiss mortality data [3]. Results for an occupational exposure scenario are shown in Table 4a. Results for a public exposure scenario, corresponding to a passive exposure of five years in a school building, is shown in Table 4b. In this latest case, only ER usually considered in the general population (1 × 10^−6^–1 × 10^−4^) are given. We chose to limit the passive exposure to 10–15-year-old individuals because the adult-based model reaches its limits in terms of prediction for younger children.

Table 4a,b shows that the current regulatory limits falls within the usual ER ranges accepted in occupational setting and in the general population for Chrysotile and, to a lesser extent, for Chrysotile mixed with Amosite. The equivalent concentrations obtained for Amosite alone or Amphiboles mixtures are markedly lower than for Chrysotile. For these fibers, the 10,000 F/m^3^ OEL value corresponds only to the upper range of ER accepted in occupational settings, while the 1000 F/m^3^ target is mostly above the range of ER accepted in the general population. In the occupational scenario equivalent concentrations for lung cancer are similar to the equivalent concentration for mesothelioma for Chrysotile mixed with Amosite (Table 4a). This is not the case in the public exposure scenario however, where the EC for mesothelioma are lower relatively to the equivalent concentration for lung cancer. The equivalent concentration for mesothelioma remained low in the general public scenario, despite the fact that duration of exposure in the occupational and general public scenarios are 40 years and five years, respectively.

## 4. Discussion

The risk assessment carried out in this study shows that the risks for the occupants of the school building are very low. This is mainly due to the limited concentration of asbestos fibers in the ceilings boards (1% in mass) and the lack of measurable background concentration observed during the normal use of the building. The field and laboratory investigation conducted allowed to rule out the hypothesis of a significant chronic air contamination in the building. Further investigations conducted to substantiate the exposure observed in April 2015, such as an analysis of meteorological events in 2015 (windy periods), or an investigation of possible maintenance or construction work in the building at the time, were inconclusive and the exact origin of the peak exposure remains unknown. Interestingly, the 5000 F/m^3^ concentration measured at the time is in the order of magnitude of the concentration estimates obtained for the fall- or the breaking of a board-events (5500–10,500 F/m^3^), which corroborates the hypothesis of an incidental contamination.

Several limitations, mostly due to the retrospective nature if the assessment are worth mentioning.

Exposure estimates are based on a limited number of recent measurements, and thus not necessarily representative of past exposures and subject to uncertainties.The dose-response models available for asbestos have been built on cohorts of adult workers highly exposed to asbestos fibers. Extrapolation to low doses in the general population is therefore a source of uncertainty.

Despite of this last limitation, the risk estimates for mesothelioma are increased in pupils when compared to adults. In the average scenario, the ER for pupils is three times higher than for the teachers (Table 3) in similar exposure conditions. This results from the early age on the exposure onset in pupils. Because of the long latency period for mesothelioma (30–40 years), the annual incidence of the disease is at its highest at a late stage of life, when the likelihood to die from other reasons is relatively high. In pupils, the incidence rate will be maximal at an earlier stage of life, when the survival rate of the general population is higher. The adjustment of the dose–response model to the life expectancy in the Swiss population, which is elevated (80.5 and 84.8 years for men and women respectively), contributed to increase this effect. This emphasizes the necessity to focus the monitoring and control of asbestos in buildings likely to accommodate young children, such as early childhood institutions, primary and middle schools, or vocational schools.

At the end of the assessment, the work was returned to parents and teachers at a public meeting. On this occasion, their representatives in the working group were able to express their views on their perception of the expertise procedure. The full expert report was published on the Institute’s website. Transparency of the procedure and stakeholder participation have restored a climate of trust and the results achieved were welcomed with relief. In light of the results, no individual investigation was deemed necessary, but we recommended mentioning to their doctor the possible exposure to asbestos in case of a respiratory problem appearing in the future. We are convinced that, without the communication efforts undertaken, the same risk assessment would have met with public mistrust and would have failed to address its legitimate concerns.

With regard to the ER usually considered as acceptable, the thresholds of 10,000 F/m^3^ for occupational exposure and 1000 F/m^3^ for the general population and passive occupational exposures are arguable. The Swiss OEL is based on the historical EPA model [10] and Hodgson’s work on chrysotile exposure [11] and does not reflect anymore the current state of knowledge. Their level of protection appears sufficient when considering exposure to Chrysotile fibers, but not for Amphiboles fibers (Amosite and Crocidolite). These results emphasize the necessity to adapt the target concentration to the asbestos fibers considered. Moreover, it appears that the detection limit of the reference method VDI 3492:2013, which is used in several countries including Switzerland, is not sufficient to cope with Amphibole fibers. A significant decrease in the threshold for Amosite will in turn require a decrease of the LOD of the current method.

## 5. Conclusions

This study allowed the quantification of health risks due to asbestos exposure in a paraprofessional and environmental setting. It confirmed the value of the participatory approach to construct relevant exposure scenarios and establish a constructive communication between exposed people, experts and policy-makers. The results of the present study have an impact that goes far beyond this specific school setting. They confirm that children have to be considered as a subgroup of a population at higher risk of developing health consequences from asbestos exposure. Moreover, they allow policy-makers to take preventive measures that apply to other similar settings and adapt exposure thresholds to the best available information. This paper shows that quantitative health risk assessments are powerful tools that give the opportunity to take evidence-based public health informed decisions for health protection.

## Figures and Tables

**Figure 1 ijerph-16-05069-f001:**
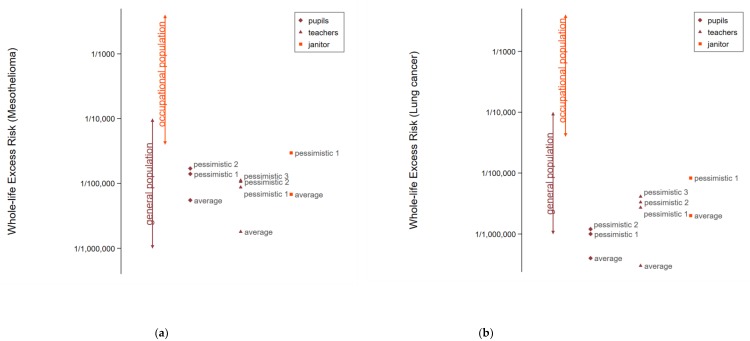
Excess risk estimates for the populations of the school (**a**) for mesothelioma (**b**) for lung cancer.

**Table 1 ijerph-16-05069-t001:** Age of exposure onset and exposure duration in the school.

Population	Scenario	First Exposure Age (yr.)	End Exposure Age (yr.)	Comment
Pupils	average	6	15	
	pessimistic 1	6	17	repetition of two years, asbestos-contaminated boards in each classroom
	pessimistic 2	6	17	scenario pessimistic 1 + unusual exposure each year (e.g., fire remediation) ^(1)^
Teachers	average	25	34	
	pessimistic 1	25	54	highest duration and weekly hours in the building among the respondents
	pessimistic 2	25	54	scenario pessimistic 1 + unusual exposure each year (e.g., fire remediation) ^(1)^
	pessimistic 3	25	65	scenario pessimistic 2 + whole worklife exposure (hypothetical scenario)
Janitor	average	30	45	
	pessimistic 1	20	60	whole worklife exposure (hypothetical scenario)

^(1)^ Since a constant exposure is assumed in the dose-response model, the exposure associated with the fire remediation was added to the average annual concentration. It corresponds to a pessimistic situation in which an event similar to this fire takes place each year.

**Table 2 ijerph-16-05069-t002:** Exposure concentrations, duration and frequency of asbestos-related events.

Description	Population				Average Scenario			Pessimistic Scenario		
Event	Pupils	Teachers	Janitor	Concentration (F/m^3^)	Event Duration (h)	Event Frequency (yr^−1^)	Contribution to the Yearly Dose^1^ (F/m^3^)	Event Duration (h)	Event Frequency (yr^−1^)	Contribution to the Yearly Dose (F/m^3^) ^(1)^
Background indoor exposure (regular use of the building)	●			75	940	0.5	18	1200	1.0	47
		●		75	582	0.5	11	1482	1.0	58
			●	75	1920	1	75	1920	1	75
Hitting a board (incident)	●	●	●	1000	4	19.0	40	4	38.0	79
Board replacement (regular maintenance)			●	50,000	0.25	2.5	16	0.25	5.0	33
Lamp replacement (regular maintenance)			●	20,000	0.17	37.5	65	0.17	75.0	130
Board replacement (regular maintenance)	●	●		400	4	0.06	0.05	4	0.13	0.10
Lamp replacement (regular maintenance)	●	●		200	4	0.94	0.39	4	1.9	0.78
Cutting/adjusting a board			●	9000	n.a	n.a	n.a	0.1	5	0.83
Breaking a board (incident)	●	●	●	5500	4	0.01	0.14	4.0	0.03	0.29
Fall (incl. breaking) of a board (incident)	●	●	●	10,500	4	0.01	0.27	4.0	0.03	0.55
Remediation work after a fire (incident)	●			1000	50.0	1.00	26.04			
		●		1000	61.8	1.00	32.16			
			●	1000	80	1.0	41.67			

^(1)^ The reference time for the computation of the yearly dose is 1920 h (40 h/week, 48 weeks/year). ● Exposed population yes/no.

**Table 3 ijerph-16-05069-t003:** Excess risk estimates for the populations of the school.

Population	Scenario	Av. Yearly Exposure conc. F/m^3^	ER Mesothelioma	ER Lung Cancer
Pupils	average	59	5.5 × 10^−6^	4.0 × 10^−7^
	pessimistic (1)	128	1.4 × 10^−5^	1.0 × 10^−6^
	pessimistic (2)	155	1.7 × 10^−5^	1.2 × 10^−6^
Teachers	Average	52	1.8 × 10^−6^	3.0 × 10^−7^
	pessimistic (1)	140	8.7 × 10^−6^	2.7 × 10^−6^
	pessimistic (2)	170	1.06 × 10^−5^	3.3 × 10^−6^
Janitor	Real	200	6.8 × 10^−6^	2.0 × 10^−6^
	pessimistic (1)	320	2.97 × 10^−5^	8.3 × 10^−6^

**Table 4 ijerph-16-05069-t004:** (**a**) Equivalent concentrations (F/m^3^) for an occupational exposure (adult, exposure age between 20 and 60, exposure duration 1920 h/year). (**b**) Equivalent concentrations (F/m^3^) for a public exposure scenario (teenager, exposure age between 10 and 15, exposure duration 1200 h/year)

ER	Mesothelioma				Lung Cancer
ER	Chrysotile only K_M_ 0.15 10^−8^	Mix (Chrysotile + Amphibole) K_M_ 1.3 10^−8^	Amosite only K_M_ 3.9 10^−8^	Amphiboles (Amosite + Crocidolite) K_M_ 7.95 10^−8^	All Fibers K_l_ 1.64 10^−2^
(**a**)					
4 × 10^−3^	1,122,832	129,578	43,211	21,186	155,262
1 × 10^−4^	27,996	3230	1077	528	3871
4 × 10^−5^	11,198	1292	431	211	1549
1 × 10^−5^	2799	323	108	53	387
1 × 10^−6^	280	32	11	5	39
(**b**)					
1 × 10^−4^	55,424	6395	2132	1046	28,814
1 × 10^−5^	5542	639	213	105	2881
1 × 10^−6^	554	64	21	10	288

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
