# Peer review of "Exposure to Amosite-Containing Ceiling Boards in a Public School in Switzerland: A Case Study"

_ijerph, 2019, doi:10.3390/ijerph16245069_

Round 1

Reviewer 1 Report

It is in many aspects a thought-provoking report of observations of exposure to fibres of asbestos material in the building(s) of a school involving students, teachers and other staff of the school during many years of observation. In my view it is not ready to be published in a scientific journal as presently drafted. It may, however, be revised by its authors in clarifications for the benefit of readers.

-  In terms of broad principles my recommendation is to introduce at the outset of report a presentation of the research questions addressed providing motives for the study and a statement of its objectives. Answers to the questions are recommended to feature also in authors conclusions and eventually in the published abstract.

-  An important element in a project stretching over long times of observation is the study population,

It needs to be described, including its relevant changes observed or during the period of study

-  An overriding question to be clarified is the exposure in the study population to fibres of asbestos.  The question arising concerns the authors intentions in regard of this comprehensive term. Is it a study focussing amosite asbestos or are also other asbestos minerals included and discussed?

-  In assessing health outcomes of exposure variables authors use for analyses the "adjusted dose-response model" described and referred to on lines 250 - 283 under rubric "3.5. Risk assessment".

The use of this model is clearly a core element for readers understanding in interpretation of analyses.

Personally I found it to be difficult reading. This is probably in part attributable to draft text using acronyms (EC = "whole life target excess risk" and also EC = "equitable concentration" ) with risk of readers´confusion. This is a critical passage explaining aspects fundamentally important to readers´ grasping authors intentions. My rec is to have this adjustment of model scrutinized in a review of report statistics.

Some additional Reviewers questions:

Reviewers Q on the study population:

I find nowhere in the manuscript a definition of the study population in the design and planning of study. The authors are likely to appre Reviewer X comments to manuscript submitted by authors Vernez D….Guillemin M on “Exposure to amosite…….a case study”      IJERPH journal manuscr 647856

Comment lines 14 – 27.

1) Ll 17-18 

Ms:  A dose response-model was adapted for amosite-only exposure and adjusted for the life expectancy and the lung cancer incidence in the Swiss population;

Reviewers Qs: What is really implied with this model? It is stated as a model adjusted for calculation of life expectancy

And the lung cancer incidence in the Swiss population. How was this done? Statement to be supported by literature reference or draft plain text to explain to readers what was actually done

2) Lines 19-20. High fibre concentrations assessed not to represent regular use of building. What is implied by regular use of a building? And what did they represent thbase in this study.en? Puzzling statement in a scientific paper abstract.

3) Lines 22-23 “Results evidenced children to be at a higher risk when exposed to asbestos”.

Is this an accurate statement of authors reflecting an evaluation of contribution to evidence base by this study under review? It is a strong statement.

Reviewers rec to revisit statement for checking of accuracy.

4) Lines 25-26 Statement in conclusion: “Quantitative health risk assessments and participatory approaches are powerful tools to support public decisions and build constructive communication tools”

This statement certainly makes sense. I fail, however, to identify presentation of its basis in “results and discussion.”

Authors recommended to revisit draft manuscript. It is of critical importance to demonstrate basis of conclusions and inferences in own study-if necessary seeking support in studies published by others.

5) Authors often use Swiss register data for reference in giving numbers and describing trends in mesothelioma incidence stretching over long time periods. Are there to be seen any trends or changes in national databases of Switzerland?

6) under rubric “context of the study” the aim of study is referred to by authors to describe a passive exposure to asbestos in a Swiss school. The presentations of investigations and measurements which have been carried out is disjointed and the reader is given a mixed impression of study population at all times during the study being exposed under variable fibre concentrations taking into account asbestos material being most  of the time materially embedded and not really being a source of exposure. On lines 63-64 is mentioned that ceiling boards containing asbestos were covering about half of the 29 classrooms and considered to be capable of releasing asbestos fibres when mechanically disturbed. On the other hand authors are aware of asbestos-containing boards having been disturbed and damaged or intervened with under circumstances making for release of friable material and exposure of people to air contaminated with asbestos fibres

7) Sections “Materials and Methods” covering text under rubrics 2.1. and 2.2 “Exposure scenarios” and 2.3. “exposure concentrations” and indeed most of draft text of page 3 of manuscript from lines 90 - 91  to line 141 is difficult reading. It is a mixture of references to scenarios implying exposures to fibres and giving glimpses of descriptions of efforts made by a working group seeking to grasp the exposures arising from being in the school building for varying lengths of time for work or studies every now and then experiencing events affecting release of friable fibrous materials.

Lines 90 – 141

The presentation of the Materials and methods section seems to me confusing

It describes the input provided by a working group charged with drafting of a report on what is known about fibre exposures in the setting of an old school building over a long stretch of years and setting the focus on endpoints lung cancer and mesothelioma, respectively.

The composition of this working group was comprehensive in seeking to include external experts such as hygienists, toxicologists and representatives of the exposed populations, school pupils, pupils parents, and still others. Steps were taken to ensure transparency process to be closely observed with the working group peers communicating with their own professional elders.

The study populations were considered to be:

1) School students aged 6 to 17 years of age and teachers; and the janitor of the school.

Duration of exposures of school pupils and school teachers with regard to weekly hours in presence school building at school times and monitoring of measurements – both as a matter of routine and those seen as being event-related in some significant way.

The two sources of exposures were consistently observed and kept apart by the working group as

Ambient exposure in principle dependent on time spent in the school building and the background indoor concentrations of pollutant materials and event-related exposures leading to contamination of indoor air with pollutants of diverse kind. All such events were to be registered and followed.  

Q; What type of asbestos was found to be most common as source of airborne fibres ?

Q: Readers of this manuscript are likely to be asking for a summarizing account of findings based on the large numbers of measurements carried out for identification of fibre types and for assessments of exposure levels.

Author Response

see attached word document

Reviewer 2 Report

The authors have presented a practical application of risk assessment used to assess concerns over asbestos exposure in a school.  The work is well-done and results are clearly presented.  An exposure assessment project was completed to better understand the range of potential exposures and in developing the risk analysis the authors were required to adjust existing inputs (dose-response metrics) to more accurately represent the type of fiber present.  I have a few concerns/clarifications that should be considered before publication:

1) I would like to see a little bit more explanation of the outreach and community collaboration.  Was a particular process followed?  Did the team regularly update the school community?

2) Please explain what amosite is, e.g., type of fiber, etc. in the Introduction.

3) I agree that the risks estimated are low but is there any external reference that can be used for comparison?  In US, EPA's "acceptable risk range" of 1 in 10,000 to 1 in 1,000,000 excess lifetime cancer risk is often used.  Is there something similar in Switzerland or at WHO?

4) Line 305-306: I disagree with the statement that adequacy of high-dose studies is arguable.  This is a common problem in risk assessment and human data, even at high-dose is useful for risk assessment purposes but likely leads to over-estimates of risk.  I suggest describing this as uncertainty (rather than adequacy) and also mention that if the results are overestimates then true risk is likely even lower (so the analysis ends up being protective of public health).

Typos/language issues: 

Line 38: use "their potencies differ" or "their potency differs"

Lines 50-52: confusing to start sentence with reference to 2009 and then describe data from 1990-93.  Maybe separate into 2 sentences or explain that there are no newer data.

Line 84: no 's' on allow

Line 119: no 's' on feedback

Line 127: "measurement was not possible"

Line 195: Start sentence with word not number "Sixteen"

Line 326: no 's' on reflect

Line 390-391: Monte-carlo simulation was used to estimate a maximal background

Line 397:replace 'a last' with "at least"

Author Response

see attached word document
